# Spatial-Temporal Evolution and Prediction of Carbon Storage: An Integrated Framework Based on the MOP–PLUS–InVEST Model and an Applied Case Study in Hangzhou, East China

Yonghua Li [1,2,3], Song Yao [1], Hezhou Jiang [1], Huarong Wang [1], Qinchuan Ran [1], Xinyun Gao [1], Xinyi Ding [1] and Dandong Ge [1,3,*]

1    Department of Regional and Urban Planning, Zhejiang University, Hangzhou 310058, China
2    Center for Balance Architecture, Zhejiang University, Hangzhou 310058, China
3    Architectural Design and Research Institute of Zhejiang University Co., Ltd., Hangzhou 310058, China
*    Correspondence: zjugdd@zju.edu.cn

**Abstract:** Land-use/land-cover change (LUCC) is an important factor affecting carbon storage. It is of great practical significance to quantify the relationship between LUCC and carbon storage for regional ecological protection and sustainable socio-economic development. In this study, we proposed an integrated framework based on multiobjective programming (MOP), the patch-level land-use simulation (PLUS) model, and the integrated valuation of ecosystem service and trade-offs (InVEST) model. First, we used the InVEST model to explore the spatial and temporal evolution characteristics of carbon storage in Hangzhou from 2000 to 2020 using land-cover data. Second, we constructed four scenarios of natural development (ND), economic development (ED), ecological protection (EP), and balanced development (BD) using the Markov chain model and MOP, and then simulated the spatial distribution of land cover in 2030 with the PLUS model. Third, the InVEST model was used to predict carbon storage in 2030. Finally, we conducted a spatial correlation of Hangzhou's carbon storage and delineated carbon storage zoning in Hangzhou. The results showed that: (1) The artificial surfaces grew significantly, while the cultivated land decreased significantly from 2000 to 2020. The overall trend was a decrease in carbon storage, and the changing areas of carbon storage were characterized by local aggregation and sporadic distribution. (2) The areas of artificial surfaces, water bodies, and shrubland will continue to increase up to 2030, while the areas of cultivated land and grassland will continue to decrease. The BD scenario can effectively achieve the multiple objectives of ecological protection and economic development. (3) The carbon storage will continue to decline up to 2030, and the EP scenario will have the highest carbon storage, which will effectively mitigate the carbon storage loss. (4) The spatial distribution of carbon storage in Hangzhou was inextricably linked to the land cover, which was characterized by a high–high concentration and a low–low concentration. The results of the study can provide decision support for the sustainable development of Hangzhou and other cities in the Yangtze River Delta region.

**Keywords:** carbon storage; land-use/land-cover change; multiobjective programming; PLUS model; InVEST model; multiple scenario simulation



## 1. Introduction

Global warming is an important environmental issue of widespread concern to the international community [1,2]. Carbon emissions are one of the causes of global warming [3], which poses a great challenge to the sustainable development of both human society and the environment [4]. Terrestrial ecosystems are important global carbon pools and play a key role in maintaining the global carbon cycle [5,6] and regulating climate change [7]. Studies showed that human-induced LUCC is a major driver of carbon storage changes in terrestrial ecosystems [8–10]. Therefore, a quantitative assessment of the corresponding

relationship between carbon storage and LUCC is of great practical significance for regional ecological protection and sustainable socio-economic development.

In recent years, many scholars have conducted numerous studies on the relationship between the role of LUCC and carbon storage and its influence mechanisms at the global [11], national [12], urban cluster [13], provincial [14], and municipal [15] scales. In terms of research content, previous studies mostly analyzed changes in carbon storage and their driving factors based on land-use/land-cover data. For example, Yin Ren et al. studied the effects of rapid urban sprawl on urban forest carbon storage [16]. Zhang et al. studied the changes in soil organic carbon following the "Grain-for-Green" Programme in China [17]. Maia et al. studied the changes in soil organic carbon storage under different agricultural management systems [18]. With the widely used InVEST model, which can estimate carbon storage easily and reliably [19–21], scholars started to combine this and land-use/land-cover simulation models, such as CA [6], CLUE-S [22], FLUS [23], and PLUS [24], to predict future carbon storage situations. For example, Shao Zhuang et al. used the FLUS–InVEST model to predict carbon storage in Beijing in 2035 under three scenarios [25]. Lin et al. used the PLUS–InVEST model to predict the distribution of land use and carbon storage in Guangdong Province in 2050 [26]. However, in terms of simulation scenario settings, most scholars focused on thinking about technical methods; for example, Cao et al. defined different scenarios by setting different transfer probabilities in the Markov chain model [27] and Liu et al. defined different scenarios by adjusting the conversion cost matrix [28]. These methods did not consider benefit targets. However, China is now facing multi-dimensional development requirements, such as new urbanization, ecological civilization, and rural revitalization. Therefore, it is necessary to consider the multiple objectives of economic, social, and ecological benefits when establishing simulation scenarios. In addition, previous studies did not sufficiently delve into the analysis of carbon storage projections, and few explored the fine-grained partitioning of carbon storage [25]. In September 2020, China put forward the goals of "carbon peaking" by 2030 and "carbon neutrality" by 2060, and the "Report on the Work of the State Council in 2021" proposed to help achieve the "double carbon" goal through territorial spatial planning [29] and other means. The Report of the State Council on the Work of the Government of the People's Republic of China in 2021 also proposed that the "double carbon" target should be achieved through territorial and spatial planning. Therefore, it is necessary to further explore possible pathways for applying the findings to planning practice when making carbon storage predictions.

Here, we proposed an integrated framework based on MOP, the PLUS model, and the InVEST model. MOP was used to deal with multiple conflicting objectives [30], the PLUS model was used to simulate the land cover spatial distribution pattern [31], and the InVEST model was used to assess and predict the spatial distribution in carbon storage [32]. This research framework was applied to Hangzhou, East China.

The aims of this study were to (1) identify the spatial and temporal evolution characteristics of carbon storage in Hangzhou from 2000 to 2020, (2) simulate the spatial distribution of land cover under four scenarios, (3) predict future carbon storage and spatial distribution patterns, and (4) delineate carbon storage zones. Overall, this study creatively combined the goals of ecological conservation and economic development with the prediction of carbon storage, and its results can provide decision support for the sustainable development of Hangzhou and other cities in the Yangtze River Delta region.

## 2. Study Area and Data

### 2.1. Study Area

Hangzhou is located in the north of Zhejiang Province, East China, (29°11′–30°34′ N, 118°20′–120°37′ E). The administrative divisions of Hangzhou are Yuhang, Gongshu, Xiacheng, Shangcheng, Jianggan, Xihu, Binjiang, Xiaoshan, Fuyang, Lin'an, Tonglu, Chun'an, and Jiande (2020), with a total area of 16,850 km² (Figure 1). Hangzhou has varied geo-

graphical conditions. In the western, central, and southern parts, there are mainly hills or mountains, and the northeastern part is mainly covered by plains.

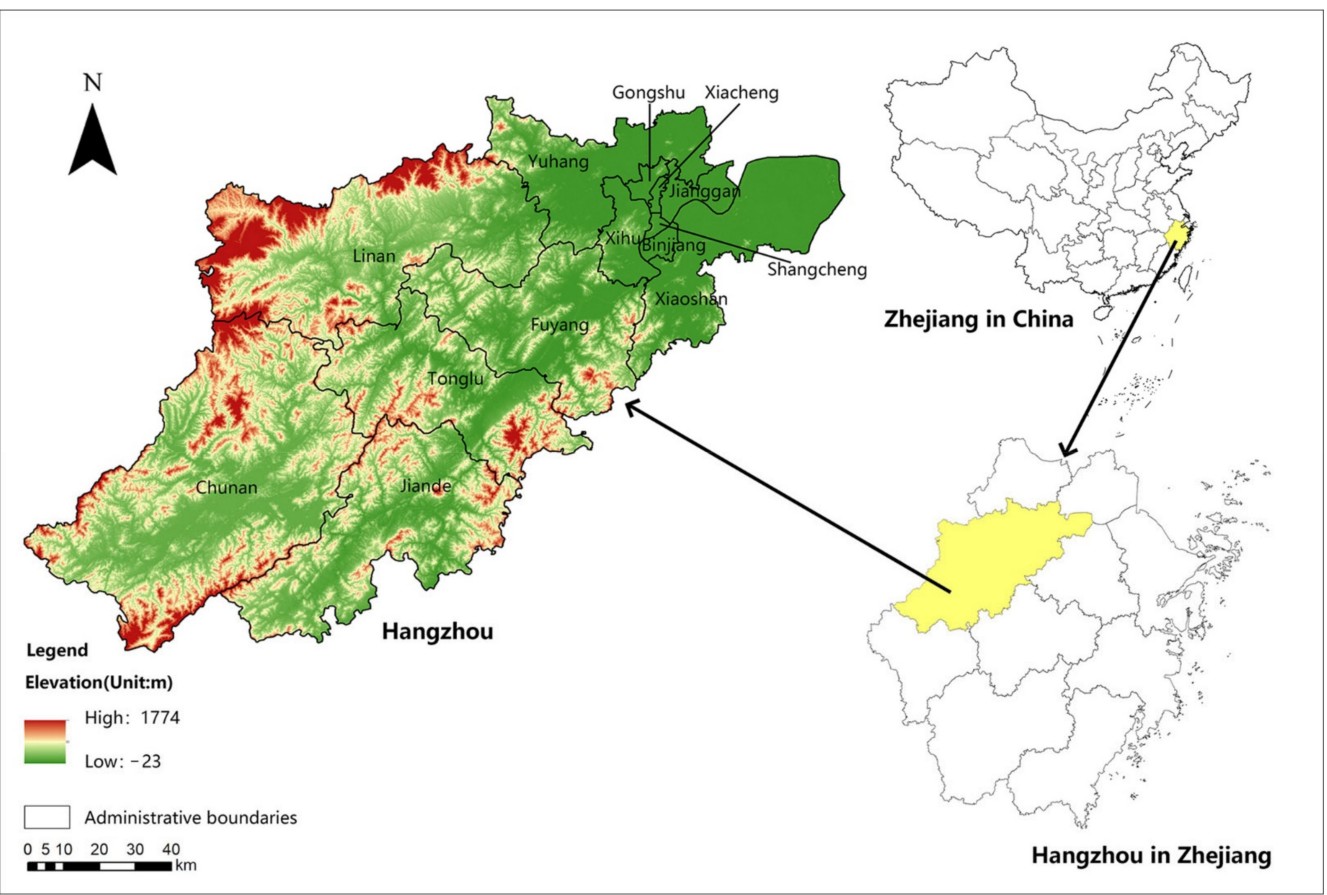

**Figure 1.** Location of Hangzhou.

As the social, economic, cultural, scientific, and educational center of Zhejiang Province, Hangzhou is focused on the multiple objectives of maintaining economic development, protecting the ecological environment, and ensuring social progress. In the past 20 years, the rapid economic and social development of Hangzhou has led to a rapid growth in artificial surfaces, which has brought about huge changes in land cover, resulting in a drastic decrease in ecosystem service functions and rapid changes in carbon storage [33,34]. Therefore, it is necessary to simulate and predict the land cover and carbon storage in Hangzhou based on the multiple objectives of future economic development and ecological protection.

*2.2. Data Sources*

The land-cover data for 2000, 2010, and 2020 used in this study were obtained from GlobeLand 30 global land-cover data, which has an overall accuracy of 85.72% and a kappa coefficient of 0.82, with high accuracy of interpretation; the land-cover map of Hangzhou was obtained after mask extraction. Seven land-cover types were identified in the study area, namely, cultivated land, forest, grassland, shrubland, wetland, water bodies, and artificial surfaces, at a spatial resolution of 30 m. The driving socio-economic factors were population density [35], GDP density [36], nighttime light [37], distance to major roads, distance to water bodies, distance to railroads, and distance to highways. Climate and environmental driving factors were the mean annual temperature, annual precipitation, the digital elevation model (DEM) [38], the normalized difference vegetation index (NDVI) [39], and the slope. The above data were unified in ArcGIS 10.2 into the WGS 1984 coordinate system and all raster data were resampled and processed to a 30 m resolution.

The carbon density data (carbon storage per unit area) [11,40] were obtained from references. The ecological and economic benefits of Hangzhou were calculated based on data from the Hangzhou Statistical Yearbooks from 2000 to 2020. The types, resolutions, and sources of data are shown in Table 1.

**Table 1.** Data information and sources.

| Category | Data | Year | Resolution | Source |
|---|---|---|---|---|
| Land cover | Land-cover data | 2000, 2010, 2020 | 30 m | http://www.globallandcover.com/ (accessed on 10 October 2022) |
| Socioeconomic factors | Population density | 2019 | 1 km | http://www.resdc.cn/ (accessed on 11 October 2022) |
| | GDP density | 2019 | 1 km | http://www.resdc.cn/ (accessed on 11 October 2022) |
| | Night light data | 2013 | 500 m | http://www.resdc.cn/ (accessed on 10 October 2022) |
| | Distance to main roads | 2020 | | https://www.openstreetmap.org (accessed on 3 August 2022) |
| | Distance to railroad | 2020 | | https://www.openstreetmap.org (accessed on 3 August 2022) |
| | Distance to highway | 2020 | | https://www.openstreetmap.org (accessed on 3 August 2022) |
| | Distance to water bodies | 2020 | | https://www.openstreetmap.org (accessed on 3 August 2022) |
| Climate and environmental factors | Annual average temperature | 2020 | 1 km | http://www.resdc.cn/ (accessed on 10 October 2022) |
| | Annual precipitation | 2020 | 1 km | http://www.resdc.cn/ (accessed on 10 October 2022) |
| | DEM | 2020 | 30 m | https://www.gscloud.cn/ (accessed on 12 October 2022) |
| | Slope | 2020 | 30 m | Retrieved from DEM |

## 3. Method

The research framework is shown in Figure 2.

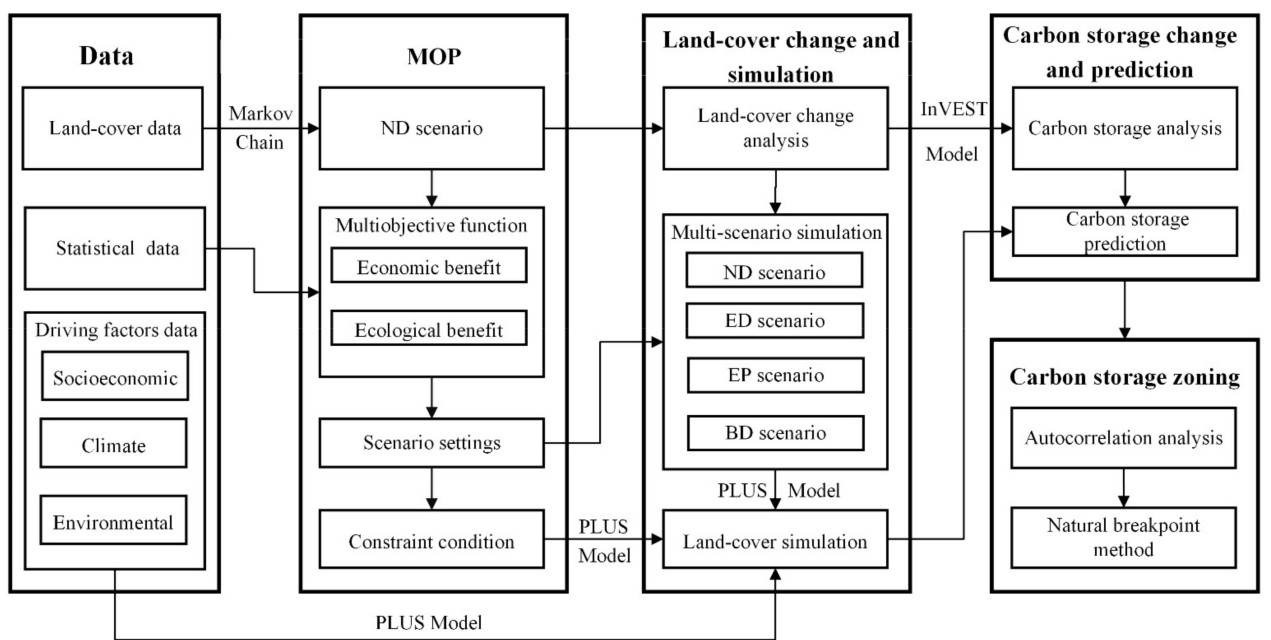

**Figure 2.** Research framework.

### 3.1. GM (1, 1) Model

The GM (1, 1) model is an important method for studying discrete data series with small numbers of samples and incomplete information [41]. The essence of the GM (1, 1)

model is to cumulatively generate an original series of a single variable and then construct a first-order linear differential equation model to obtain a fitted curve to predict the system [42]. This is shown in Equation (1), where $x^{(0)}$ is the original sequence. Equation (2) presents the constructed first-order linear differential equation, where $x^{(1)}$ is the new sequence after the accumulation of the original sequence. The predicted value of the original sequence can be obtained from Equation (3).

$$x^{(1)}(i) = \sum_{i=1}^{n} x^{(0)}(i)(i = 1, 2, \ldots, n) \tag{1}$$

$$u = \frac{dx^{(1)}}{dt} + ax^{(1)} \tag{2}$$

$$x^{(0)}(i+1) = x^{(1)}(i+1) - x^{(0)}(i)(i = 1, 2, \ldots, n) \tag{3}$$

*3.2. MOP*

MOP is an important technique for land-use optimization research [43] that has two parts—the objective function and constraints—and its research results are of reference value for understanding future land-use structures and contemporary land planning [44,45]. The formulae used in the MOP are shown below: Equation (4) is the objective function, where $F_i(x)$ is the ith objective function, $x_j$ is the jth decision variable, and $a_j$ is the corresponding decision coefficient; Equation (5) is the constraint condition, where $b_{ij}$ is the coefficient corresponding to the jth variable in the ith constraint condition, and $c_j$ is the corresponding constraint value.

$$maxF_i(x) = \sum_{j=1}^{n} a_j x_j, (i, j = 1, 2, \ldots, n) \tag{4}$$

$$s.t. \begin{cases} \sum_{j=1}^{n} b_{ij} x_j = (\geq, \leq) c_j, (i = 1, 2, \ldots, n) \\ x_j \geq 0, (j = 1, 2, \ldots, n) \end{cases} \tag{5}$$

3.2.1. Construction of the Multiobjective Function

The contemporary characteristics and development of land resources in Hangzhou are mainly a result of the combined influence of social, economic, and ecological benefits; however, because social benefits mainly refer to the degree of demand for land by various sectors of society, which is mainly reflected in the restrictions or protection of various land-cover types in relevant policy documents, it is difficult to quantify with a single maximum or minimum objective function [46]. Thus, this study selected economic benefits and ecological benefits as the two direct objectives, and social benefits were reflected in the constraints.

(1)   Ecological benefit objectives

Based on the total grain PV and the area of cultivated land in each year from 2000 to 2020 in the statistical yearbooks for Hangzhou, the average output value of grains per unit area in Hangzhou in the previous ten years was calculated to be 19,240.99 CNY/hm$^2$/year. The ecological benefit values of land-cover types in Hangzhou were then measured based on China's ecosystem service value equivalent per unit area proposed by Xie et al. [47]; the results are shown in Table 2.

(2)   Economic benefit objective

The total agricultural output value, sub-production value, and output value of two or three industries in each year from 2000 to 2020 were obtained from the statistical yearbooks of Hangzhou disclosed by Hangzhou's government, and the economic efficiency value of Hangzhou from 2000 to 2020 was calculated according to the related formula; then the values of each land cover type in 2030 were predicted by using the GM (1, 1) model. The results are shown in Table 3.

**Table 2.** Ecological efficiency factors of each land-cover type in Hangzhou.

| Land-Cover Type (A) | Ecological Factor | Ecological Efficiency Value (Unit: $10^4$ CNY/hm$^2$/year) (CNY—Chinese Yuan) |
|---|---|---|
| Cultivated land (A1) | Dryland + paddy | 1.11 |
| Forest (A2) | Coniferous forest + mixed coniferous forest + broadleaf forest | 7.91 |
| Grassland (A3) | Grassland + scrub + meadow | 1.89 |
| Shrubland (A4) | Shrubs | 4.50 |
| Wetland (A5) | Wetland | 6.47 |
| Water bodies (A6) | Water system | 15.62 |
| Artificial surfaces (A7) | Urban and rural construction land | 0.02 |

**Table 3.** Economic efficiency factors of each land-cover type in Hangzhou.

| Land-Cover Type (B) | Formula (PV: Production Value) | Economic Efficiency Value in 2030 (Unit: $10^4$ CNY/hm$^2$/year) (CNY—Chinese Yuan) |
|---|---|---|
| Cultivated land (B1) | (Agricultural PV − tea fruit PV)/cultivated land area | 10.23 |
| Forest (B2) | Forestry PV/forest land area | 1.06 |
| Grassland (B3) | Pastoral PV/grassland area | 11.57 |
| Shrubland (B4) | Tea fruit PV/shrubland area | 1044.26 |
| Wetland (B5) | Fishery PV/(water bodies area + wetland area) | 0.12 |
| Water bodies (B6) | | 5.46 |
| Artificial surfaces (B7) | (Secondary industry PV + tertiary industry PV)/artificial surfaces area | 1762.60 |

### 3.2.2. Scenario Setting

Four scenarios were developed for this study: natural development scenario (ND), economic development priority (ED), ecological protection priority (EP), and balanced development (BD). The ND scenario followed the natural evolution of land-cover types without additional constraints. The area of each land-cover type under this scenario was determined by entering the land-cover data in the study area in 2010 and 2020 in the Markov chain module of the PLUS model. The ED scenario strengthened economic development and urbanization, with the growth of economic benefits as the main objective. The EP scenario strengthened the protection of ecological land, with the growth of ecological benefits as the main objective. The BD scenario strengthened the degree of comprehensive use of land resources and promoted economic and urbanization under the premise of ensuring the sustainable development of the ecological environment, with the growth of both ecological and economic benefits as the main objective. The weights of ecological and economic benefits in the ED, EP, and BD scenarios were determined using the Delphi method [45,48] (Table 4).

**Table 4.** Scenario setting.

| Scenario | Economic Efficiency Target Weighting | Ecological Efficiency Target Weighting |
|---|---|---|
| Economic development (ED) | 0.80 | 0.20 |
| Ecological protection (EP) | 0.20 | 0.80 |
| Balanced development (BD) | 0.50 | 0.50 |

### 3.2.3. Constraints

To ensure that the future land-cover changes in Hangzhou under multiple objectives and scenarios were consistent with the laws of social development and relevant macro policy expectations, this study integrated the land-cover data under the ND, the natural decay rate of land cover in Hangzhou from 2010 to 2020, the objectives and requirements of the relevant 14th Five-Year Plan documents issued by Hangzhou, and the coordination

relationship between each scenario and social benefits needs. The results are shown in Table 5.

**Table 5.** Multiobjective function constraints.

| Constraint Factors | Constraint Range (hm$^2$) | Description |
|---|---|---|
| Total constraint | $X_0$ = 1,685,639.88 | Ecological space, living space, and production space. The total area of these three types of land space constituted the total area constraint. |
| Cultivated land constraint | 295,485.12 $\geq X_1 \geq$ 282,675.10 | Considering China's cultivated land protection policy, the rate of decrease in the cultivated land area from 2020 to 2030 should be no higher than the rate of decrease in the cultivated land area from 2010 to 2020. Thus, the rate of decrease from 2010 to 2020 was taken to measure the lower limit constraint of the cultivated land area from 2020 to 2030. Additionally, the cultivated land area in 2030 under the ND scenario was taken as the upper limit constraint. |
| Forest constraint | (ED) 1,021,124.52 $\geq X_2 \geq$ 687,464.02 | According to the "14th Five-Year Plan for Forestry Development" for Hangzhou in 2021, the target forest coverage should be greater than 66.8% of that in 2020. According to the policy, in the ED scenario, 66.8% of the current forest area in 2020 was taken as the lower limit constraint. Additionally, the forest area in 2030 under the ND scenario was taken as the upper limit constraint. |
| | (EP and BD) $X_2 \geq$ 687,464.02 | In the EP and BD scenarios, the forest should be strictly protected; therefore, the current forest area in 2020 was taken as the lower limit constraint. |
| Grassland constraint | (ED) $X_3 \leq$ 64,775.79 | In the ED scenario, economic development should be a priority; therefore, the current grassland area in 2020 was taken as the upper limit constraint. |
| | (EP and BD) 65,689.22 $\geq X_3 \geq$ 59,433.10 | A 5% increase in the grassland area in 2030 under the ND scenario was taken as the upper limit constraint, and a 5% decrease was taken as the lower limit constraint. |
| Shrubland constraint | (ED) $X_4 \leq$ 1692.18 | In the ED scenario, economic development should be a priority; thus, the shrubland area in 2030 under the ND scenario was taken as the upper limit constraint. |
| | (EP and BD) 1861.40 $\geq X_4 \geq$ 1625.76 | A 10% increase in the shrubland area in 2030 under the ND scenario was taken as the upper limit constraint. The current shrubland area in 2020 was taken as the lower limit constraint. |
| Wetland constraint | (ED) 2982.42 $\geq X_5 \geq$ 1548.90 | According to the "14th Five-Year Plan for Wetland Protection" for Hangzhou in 2021, the wetland area in 2025 was targeted to reach at least 55% of that in 2020. According to the policy, in the ED scenario, 55% of the current wetland area in 2020 was taken as the lower limit constraint. Additionally, the wetland area in 2030 under the ND scenario was taken as the upper limit constraint. |
| | (EP and BD) $X_5 \geq$ 2816.19 | In the EP and BD scenarios, the wetland should be strictly protected; therefore, the current wetland area in 2020 was taken as the lower limit constraint. |

**Table 5.** *Cont.*

| Constraint Factors | Constraint Range (hm²) | Description |
|---|---|---|
| Water bodies constraint | (ED)<br>$116{,}521.56 \geq X_6 \geq 95{,}324.66$ | According to the "14th Five-Year Plan for Water & Soil Conservation" and the "14th Five-Year Plan for Water Ecological Protection" for Hangzhou in 2021, the main bodies of watersheds should be protected, and the soil and water bodies area in 2025 should reach at least 94% of that in 2020.<br>According to the policy, in the ED scenario, 94% of the current water bodies area in 2020 was taken as the lower limit constraint. Additionally, the water bodies area in 2030 under the ND scenario was taken as the upper limit constraint. |
| | (EP and BD)<br>$X_6 \geq 101{,}409.21$ | In the EP and BD scenarios, water bodies should be strictly protected; thus, the current area of water bodies in 2020 was taken as the lower limit constraint. |
| Artificial surfaces area constraint | (ED)<br>$X_7 \leq 203{,}800.21$ | In the ED scenario, a 10% increase in the artificial surfaces area in 2030 under the ND scenario was taken as the upper limit constraint. |
| | (EP)<br>$X_7 \leq 166{,}745.63$ | In the BD scenario, a 5% increase in the artificial surfaces area in 2030 under the ND scenario was taken as the upper limit constraint and a 5% decrease was taken as the lower limit constraint. |
| | (BD)<br>$194{,}536.57 \geq X_7 \geq 176{,}009.27$ | In the BD scenario, a 5% increase in the artificial surfaces area in 2030 under the ND scenario was taken as the upper limit constraint and a 5% decrease was taken as the lower limit constraint. |

### 3.3. The PLUS Model

The PLUS model was developed by the High-Performance Spatial Computing Intelligence Laboratory of China University of Geosciences (Wuhan) [49,50]. The model integrates the Land Expansion Analysis Strategy (LEAS) and a CA model based on muti-type random patch seeds (CARS) to explore the drivers of land expansion and landscape change. Compared with other models, the PLUS model can obtain higher simulation accuracy and more similar landscapes [50].

The specific parameters of the PLUS model in this study were set as follows: in the LEAS module, the number of regression trees = 50, mTry = 5, and sampling rate = 0.01. In the CARS module, neighborhood size = 3, patch generation = 0.9, expansion coefficient = 0.1, and thread = 0.9. In the CARS (CA based on multiple path seeds) module, neighborhood size = 3, patch generation = 0.9, expansion coefficient = 0.1, and thread = 0.9. The neighborhood weights for each land-cover type were calculated by calculating the expansion area of land cover type as a percentage of the total land-cover expansion area. The final parameters were 0.196511, 0.293085, 0.134640, 0.003633, 0.005221, 0.116621, and 0.250289.

Model accuracy validation was needed before the simulation of the 2030 land cover. First, we loaded the land-cover data of 2010 and 2020 in the Extract Land expansion module to obtain the land expansion map from 2000 to 2010; second, we loaded the Land Expansion map from 2000 to 2010 and the driving factors into the LEAS module to obtain the development potential of each category during this period; finally, we loaded the land-cover data for 2020 and the development potential of each type in the CARS module to obtain the simulated land cover in 2020 and compared them with the real value for 2020. The overall accuracy of the simulation reached 85.51%, the kappa value was 0.75, and the figure of merit (FOM) value was 0.15, which met the requirements.

### 3.4. The InVEST Model

The InVEST model is an ecosystem service assessment model [15], which is a tool for exploring how changes in ecosystems are likely to lead to changes in benefits that flow to people [51]. It can estimate the amount of carbon sequestration with maps of land-cover types and data on carbon storage in four carbon pools (i.e., aboveground biomass, belowground biomass, soil, and dead organic matter). The basic assumption of the module is that the carbon density of a given land type is considered to be a constant and does not change over time [52]. The calculation formula is as follows:

$$C_k = C_{k\_above} + C_{k\_below} + C_{k\_soil} + C_{k\_dead} \tag{6}$$

$$C_{Toal} = \sum_{k=1}^{n} A_k \times C_k \tag{7}$$

where $k$ is the kth land cover type, $C_k$ is the kth total carbon density (t/hm$^2$), $C_{k\_above}$ is the kth aboveground vegetation carbon density, $C_{k\_below}$ is the kth belowground vegetation carbon density, $C_{k\_soil}$ is the kth soil carbon density, and $C_{k\_dead}$ is the kth dead organic matter carbon density. $C_{Total}$ is the total carbon storage (t), n is the total number of land-use types, and $A_k$ is the area of kth land-cover type (hm$^2$).

The carbon density has a considerable influence on the results of the total carbon storage in the study area [53]. Ideally, the carbon storage of the landscape should be determined by large-scale soil sampling [54]. This is not always feasible due to the long time and high cost constraints; therefore, many scholars use the carbon density data source from the extant literature [54,55].

In this study, we derived the carbon density data of each land-cover type (Table 6) from the references available for China [56–62]. The carbon density data from different studies vary considerably and the spatial patterns of carbon density are strongly correlated with climatic variables [63,64]; thus, we developed principles for the selection of carbon density data: we prioritized carbon density data from Zhejiang Province, followed by carbon density data from areas in the same climate zone (subtropical monsoon) as Hangzhou. Sources of carbon intensity data for different land covers are shown in Table 6.

**Table 6.** Carbon density of each land-cover type (unit: t/hm$^2$).

| Land-Cover Type | $C_{k\_above}$ | $C_{k\_below}$ | $C_{k\_soil}$ | $C_{k\_dead}$ | $C_k$ | References |
|---|---|---|---|---|---|---|
| Cultivated land | 16.49 | 10.89 | 75.82 | 2.11 | 105.31 | [56–58] |
| Forest | 22.62 | 18.03 | 126.75 | 2.78 | 170.18 | [56,59] |
| Grassland | 14.29 | 17.15 | 87.05 | 7.28 | 125.77 | [56,58] |
| Shrubland | 8.10 | 1.62 | 91.70 | 3.48 | 104.90 | [60] |
| Wetland | 7.40 | 24.30 | 247.80 | 1.24 | 280.74 | [61] |
| Water bodies | 1.59 | 0.00 | 64.03 | 3.98 | 69.60 | [59] |
| Artificial surfaces | 0.83 | 0.08 | 43.71 | 0.00 | 44.62 | [58,59,62] |

### 3.5. Spatial Correlation Analysis

The purpose of spatial autocorrelation analysis is to determine whether a variable is spatially correlated and to what extent it is correlated [65–67]. In this study, the spatial autocorrelation analysis was performed on Hangzhou's predicted carbon storage results in 2030. First, the create fishnet was used on ArcMap to generate 1 km×1 km grids of Hangzhou; then. Zonal Statistics as Table was used to determine the carbon storage value of each 1 km×1 km grid point. Secondly, global spatial autocorrelation analysis was performed on Geoda. Finally, we used LISA clustering maps to explore the local spatial distribution patterns.

## 4. Results

### 4.1. Land-Cover Change from 2000 to 2020

From the spatial distribution map of land cover in Hangzhou for 2000, 2010, and 2020 (Figure 3), the artificial surfaces were clustered in the northeast, which was the main urban area of Hangzhou. There was a large water body area in the southwest, which was Qiandao Lake, a national 5A-level tourist attraction in Hangzhou. As presented in Table 7, the land cover of Hangzhou was dominated by cultivated land and forest, of which forest accounted for over 60% and the cultivated land accounted for over 20%.

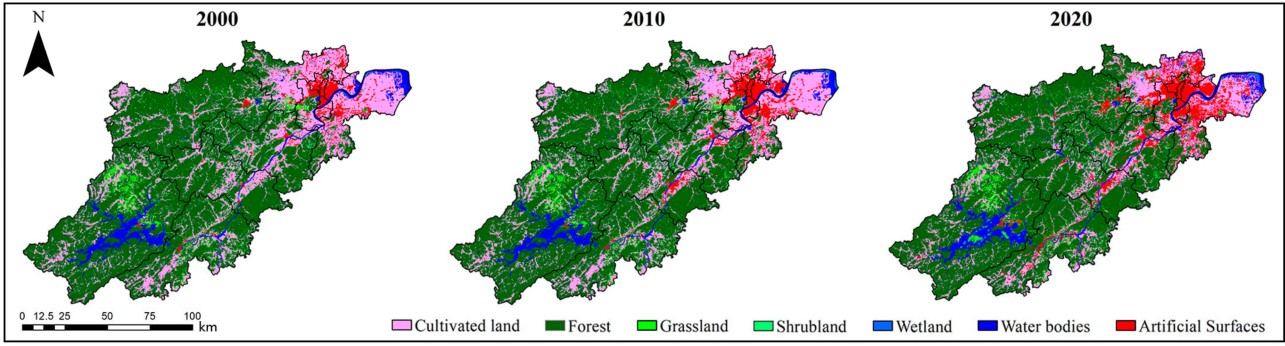

**Figure 3.** Spatial distribution of land cover in Hangzhou from 2000 to 2020.

**Table 7.** Area changes in each land-cover type in Hangzhou from 2000 to 2020.

| Land-Cover Type | Area (km$^2$) | | | Rate of Change (%) | | |
|---|---|---|---|---|---|---|
| | 2000 | 2010 | 2020 | 2000–2010 | 2010–2020 | 2000–2020 |
| Cultivated land | 4345.23 | 4122.44 | 3413.67 | −5.13 | −17.19 | −21.44 |
| Forest | 10,333.40 | 10,291.91 | 10,291.38 | −0.40 | −0.01 | −0.41 |
| Grassland | 674.25 | 682.50 | 647.76 | 1.22 | −5.09 | −3.93 |
| Shrubland | 10.68 | 14.80 | 16.26 | 38.53 | 9.86 | 52.19 |
| Wetland | 13.90 | 29.58 | 28.16 | 112.75 | −4.80 | 102.54 |
| Water bodies | 880.73 | 816.32 | 1014.09 | −7.31 | 24.23 | 15.14 |
| Artificial surfaces | 597.78 | 898.85 | 1445.09 | 50.36 | 60.77 | 141.74 |

In terms of the rate of change in each land-cover type from 2000 to 2020, the artificial surfaces had the highest rate of 141.74%, which indicated the rapid expansion of the urban and rural areas in Hangzhou over the past 20 years. Notably, wetlands reached 102.54% from 2000 to 2020, which may be attributed to the implementation of the Wetland Protection Plan in Hangzhou. The cultivated land was not well protected and was significantly reduced by 21.44% of its area. The rates of change in the forest and grassland declined slightly by 0.41% and 3.93%, respectively.

### 4.2. Spatial-Temporal Evolution of Carbon Storage from 2000 to 2020

To study the general trend in carbon storage in Hangzhou from 2000 to 2020, we used the carbon storage module of the InVEST model to estimate the data (Figures 4 and 5). As shown in Table 8, the total values of carbon storage in Hangzhou were $239.39 \times 10^6$ t, $237.75 \times 10^6$ t, and $233.69 \times 10^6$ t in 2000, 2010, and 2020, respectively, with a decreasing trend and a cumulative loss of $5.70 \times 10^6$ t. Specifically, the carbon storage decreased by only $1.6 \times 10^6$ t from 2000 to 2010, which was mainly attributed to an increase in the wetland area of 112.75%, which played an important role as a carbon sink. Moreover, carbon storage decreased sharply from 2010 to 2020 by $3.45 \times 10^6$ t and the loss of soil carbon storage was the highest by $2.05 \times 10^6$ t. During this period, the demand for economic development and urban construction accelerated, resulting in a drastic reduction in Hangzhou's carbon storage and an increase in carbon loss.

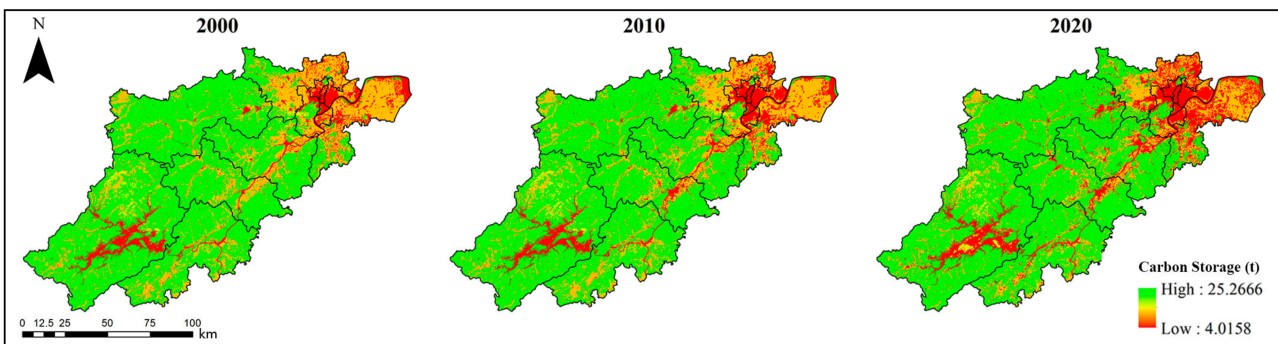

**Figure 4.** Spatial distribution of carbon storage in Hangzhou from 2000 to 2020.

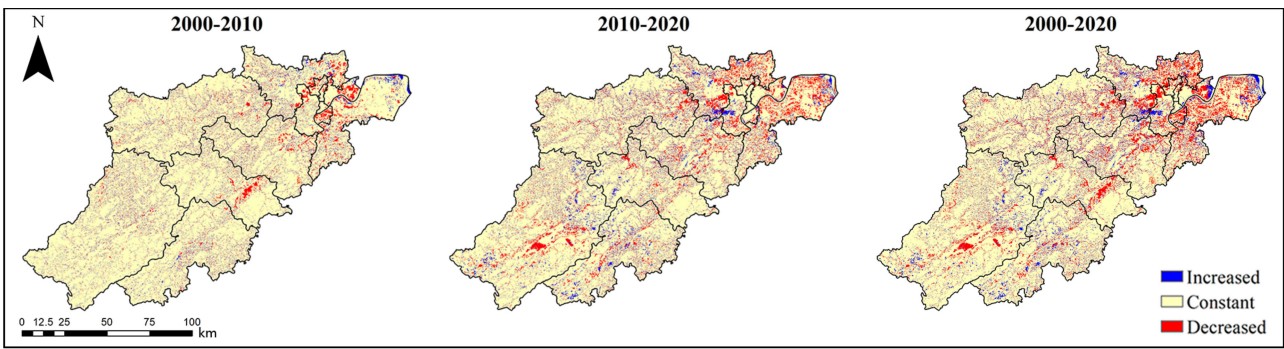

**Figure 5.** Spatial variation in carbon storage in Hangzhou from 2000 to 2020.

**Table 8.** Carbon storage in Hangzhou from 2000 to 2020 (unit: $10^6$ t).

| Year | $C_{Total\_above}$ | $C_{Total\_below}$ | $C_{Total\_soil}$ | $C_{Total\_dead}$ | $C_{Total}$ |
|------|------|------|------|------|------|
| 2000 | 31.71 | 24.56 | 178.49 | 4.64 | 239.39 |
| 2010 | 31.29 | 24.22 | 177.67 | 4.56 | 237.75 |
| 2020 | 30.15 | 23.45 | 175.62 | 4.47 | 233.69 |

The carbon storage of each land-cover type in Hangzhou changed from 2000 to 2020 (Table 9). The carbon storage of cultivated land decreased significantly from $457.60 \times 10^5$ t in 2000 to $359.49 \times 10^5$ t in 2020. The carbon storage of the forest, as the most important carbon pool, remained almost unchanged. The carbon storage of grassland rose and then fell from 2000 to 2020, reducing by $3.33 \times 10^5$ t. The carbon storage in the shrubland, wetland, and water bodies increased because of the government-implemented environmental protection for wetland parks, national nature reserves, and ecological restoration project during this period. In addition, the carbon storage of artificial surfaces increased significantly by $37.69 \times 10^5$ t from 2000 to 2020.

**Table 9.** Carbon storage of each land-cover type in Hangzhou from 2000 to 2020 (unit: $10^5$ t).

| Land-Cover Type | 2000 | 2010 | 2020 |
|------|------|------|------|
| Cultivated land | 457.60 | 434.13 | 359.49 |
| Forest | 1758.54 | 1751.48 | 1751.39 |
| Grassland | 84.80 | 85.84 | 81.47 |
| Shrubland | 1.12 | 1.55 | 1.71 |
| Wetland | 3.90 | 7.63 | 7.91 |
| Water bodies | 61.30 | 56.82 | 70.58 |
| Artificial surfaces | 26.67 | 40.03 | 64.36 |

### 4.3. Simulation of Land Cover in 2030

The spatial distribution of land cover in Hangzhou under four scenarios in 2030 was obtained by applying the PLUS model with the data of the current land-cover status and the driving factors in 2020 (Figure 6). The area of each land-cover type under the four scenarios is shown in Table 10, and the areas of artificial surfaces, water bodies, and shrubland will continue to increase, whereas those of cultivated land and grassland will continue to decrease. Specifically, under the ND scenario, the land cover will continue the same changing trend as that from the previous two decades. Additionally, the area of artificial surfaces will increase to 1852.73 km$^2$, with some land-cover expansion in Xiaoshan, Yuhang, Xihu, Fuyang, and Jiande. Under the ED scenario, the area of artificial surfaces will further increase to 2038.00 km$^2$. Compared with the ND scenario, more expansion will occur in Chun'an, Tonglu, and Lin'an. Under the EP scenario, artificial surfaces will expand slightly. Compared with the ND and ED scenarios, the artificial surfaces will decrease by 185.27 km$^2$ and 370.55 km$^2$ by 2030, respectively. In addition, the changing trend of the forest will be reversed from decreasing to increasing as the ecological land covers are protected. Finally, under the BD scenario, the area of artificial surfaces will be 1945.37 km$^2$, compared with the ND scenario, with only an obvious expansion in Tonglu.

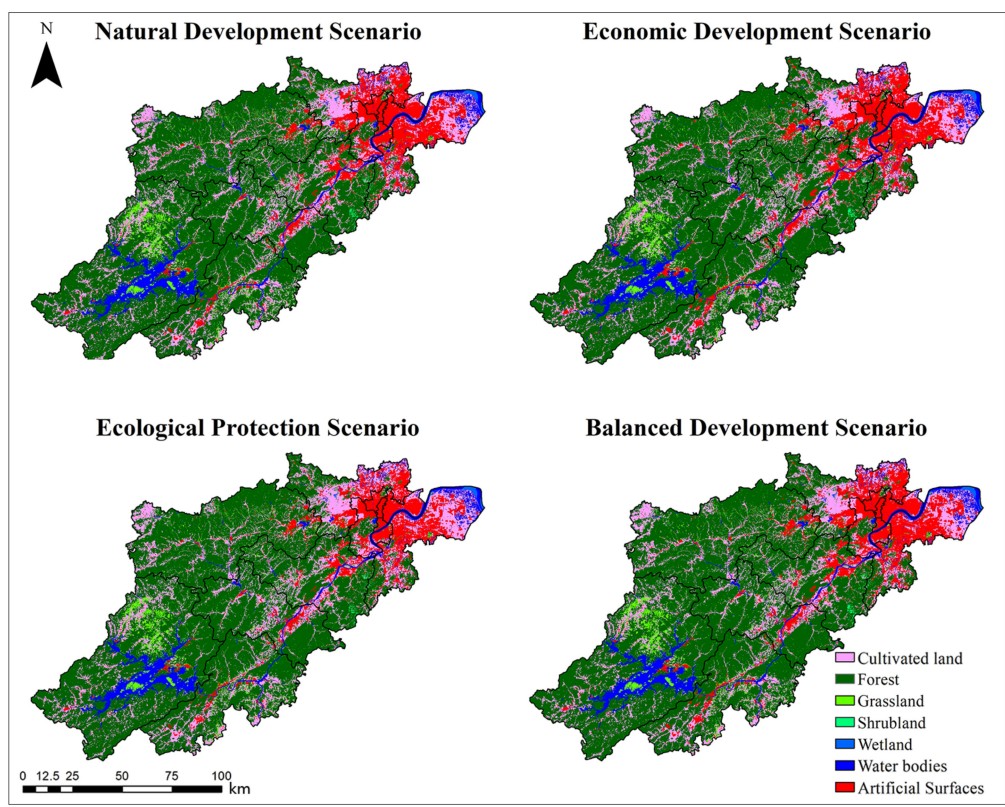

**Figure 6.** Simulation of the land-cover distribution in Hangzhou under the four scenarios in 2030.

### 4.4. Comparative Analysis of Benefits in 2030

The economic and ecological PVs in 2030 under four scenarios are shown in Table 11. The economic PV under the ND scenario will be CNY 33,379.47 × 10$^4$ and the ecological PV will be CNY 1037.11 × 10$^4$, whereas the economic PV under the ED scenario will be the highest at CNY 36,627.01 × 10$^4$ and the ecological PV will only be CNY 1034.00 × 10$^4$, which will be the lowest among the four scenarios. Under the EP scenario, the ecological PV will be the highest at CNY 1071.26 × 10$^4$, but the economic benefit will be the lowest among the four scenarios at CNY 30,123.58 × 10$^4$, which was attributed to the higher focus on the protection of ecological land cover, although it may hinder economic development. Under the BD scenario, compared with the ND scenario, the economic and ecological

PVs will increase by 4.91% and 1.18%, respectively. The BD scenario would guarantee the ecological protection of Hangzhou while ensuring reasonable economic development. It can effectively achieve both targets of the ED and EP scenarios.

**Table 10.** Area of each land-cover type in Hangzhou under four scenarios in 2030 (unit: $km^2$).

| Land-Cover Type | ND | ED | EP | BD |
|---|---|---|---|---|
| Cultivated land | 2954.85 | 2826.75 | 2826.75 | 2826.75 |
| Forest | 10,211.25 | 10,211.25 | 10,439.35 | 10,161.44 |
| Grassland | 625.61 | 582.77 | 594.33 | 594.33 |
| Shrubland | 16.92 | 16.92 | 18.61 | 18.61 |
| Wetland | 29.82 | 15.49 | 28.16 | 28.16 |
| Water bodies | 1165.22 | 1165.22 | 1281.74 | 1281.74 |
| Artificial surfaces | 1852.73 | 2038.00 | 1667.46 | 1945.37 |

**Table 11.** Comparative analysis of benefits under four scenarios in 2030.

| | ND | ED | EP | BD |
|---|---|---|---|---|
| Economic PV (billion CNY) | 33,379.47 | 36,620.88 | 30,123.58 | 35,019.06 |
| Ecological PV (billion CNY) | 1037.11 | 1018 | 1071.26 | 1049.35 |
| Percentage change in economic PV | 0.00% | 9.71% | −9.75% | 4.91% |
| Percentage change in ecological PV | 0.00% | −1.84% | 3.29% | 1.18% |

*4.5. Prediction of Carbon Storage in 2030*

Importing the land-cover simulation data under the ND, ED, EP, and BD scenario outputs from the PLUS model into the carbon storage module of InVEST model, we determined the spatial distribution of 2030 carbon storage in Hangzhou (Figure 7). In general, there were reductions in the carbon storage of aboveground, underground root, soil, and dead organic matter under four scenarios compared with the values in 2020 (Table 12); Figure 8 shows that the decreasing carbon storage was spatially concentrated in the Yuhang, Fuyang, Xihu, Xiaoshan, and Jianggan districts, while the Shangcheng, Xiacheng, and Gongshu districts exhibited no more carbon storage because they were closer to the urban center and were constructed earlier, meaning there was no space for any more expansion. Specifically, in the ND scenario, the carbon storage of the aboveground, underground root, soil, dead organic matter, and accumulative carbon storage will be $29.24 \times 10^6$ t, $22.78 \times 10^6$ t, $173.73 \times 10^6$ t, $4.39 \times 10^6$ t, and $230.14 \times 10^6$ t, respectively. In the ED scenario, the carbon storage will be the lowest at only $228.67 \times 10^6$ t. Under this scenario, in addition to the three districts near the urban center, there was no more space for further expansion in the other ten districts and the carbon storage there will reduce obviously. In the EP scenario, the carbon storage will reach $232.23 \times 10^6$ t, which will be $2.10 \times 10^6$ t and $3.56 \times 10^6$ t higher than the ND and ED scenarios, respectively. The expansion of artificial surfaces in the EP scenario will be strictly limited and it is prohibited to reduce the area of the water bodies, wetlands, and forests, effectively slowing down the decline in carbon storage. Finally, in the BD scenario, the level of carbon storage will be between the ND scenario and the EP scenario at $228.74 \times 10^6$ t.

*4.6. Spatial Correlation Analysis of Carbon Storage in 2030*

In terms of global spatial correlation, the spatial Moran's I values of carbon storage in Hangzhou under the ND, ED, EP, and BD scenarios for 2030 were all greater than 0, i.e., 0.711, 0.713, 0.708, and 0.714, respectively. Figure 9 shows the Moran's I scatter plot, in which most of the areas were located in the first quadrant (hot-spot area) and the third quadrant (cold-spot area), indicating that most spaces in Hangzhou exhibited a strong positive spatial correlation. The global spatial correlation of Hangzhou carbon storage was characterized by the fact that high-value areas tended to be adjacent to high-value areas, whereas low-value areas tended to be adjacent to low-value areas.

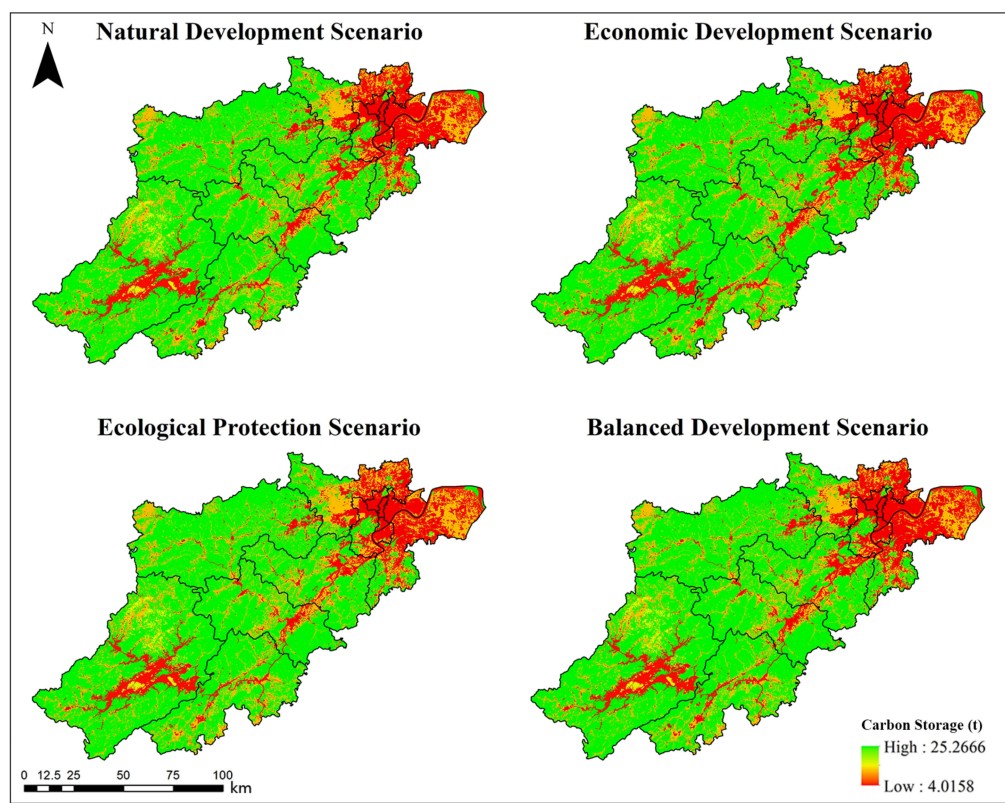

**Figure 7.** Prediction of the carbon storage distribution in Hangzhou under the four scenarios in 2030.

**Table 12.** Carbon storage in Hangzhou under four scenarios in 2030 (unit: $10^6$ t).

| Scenario | $C_{Total\_above}$ | $C_{Total\_below}$ | $C_{Total\_soil}$ | $C_{Total\_dead}$ | $C_{Total\_total}$ |
|----------|----------|----------|---------|---------|---------|
| ND | 29.24 | 22.78 | 173.73 | 4.39 | 230.14 |
| ED | 28.97 | 22.53 | 172.84 | 4.33 | 228.67 |
| EP | 29.50 | 22.99 | 175.29 | 4.45 | 232.23 |
| BD | 28.90 | 22.49 | 172.98 | 4.37 | 228.74 |

In terms of local spatial correlation, Hangzhou's carbon storage values under the four scenarios in 2030 were similar to the spatial distribution. The LISA agglomeration analysis (Figure 10) showed that the high–high carbon storage agglomeration areas and hotspot areas were mainly distributed in the northern, central, southern, and southeastern areas of the city, exhibiting a concentrated distribution trend, probably because these districts had fewer artificial surfaces and more forests. The low–low carbon storage concentration areas and cold spot areas were mainly distributed in the northeastern and southwestern areas of the city, including the main urban districts, in which more artificial surfaces and more water bodies exist. Overall, the spatial distribution of carbon storage in Hangzhou was inextricably linked to the land cover. The high values of carbon storage were distributed in the central and northern areas where there were few artificial surfaces but more ecological land cover, whereas the low-value areas were distributed in the northeastern districts where the intensity of artificial surfaces was high and the areas of ecological land cover were fragmented.

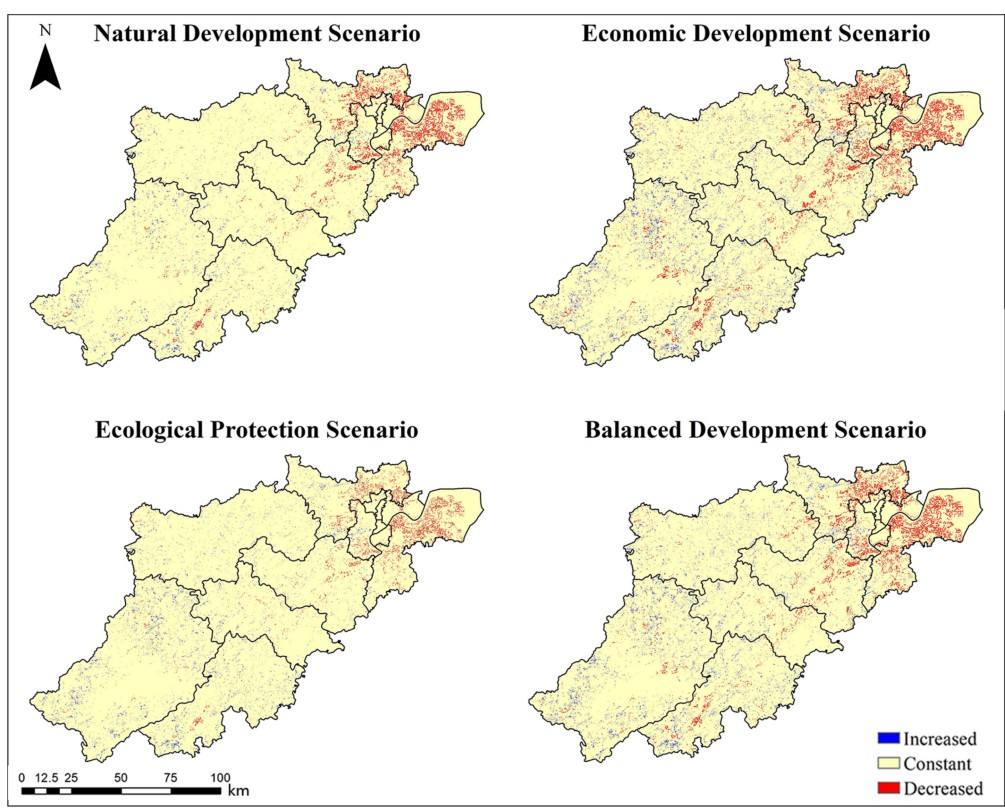

**Figure 8.** Prediction of the carbon storage change in Hangzhou under the four scenarios in 2030.

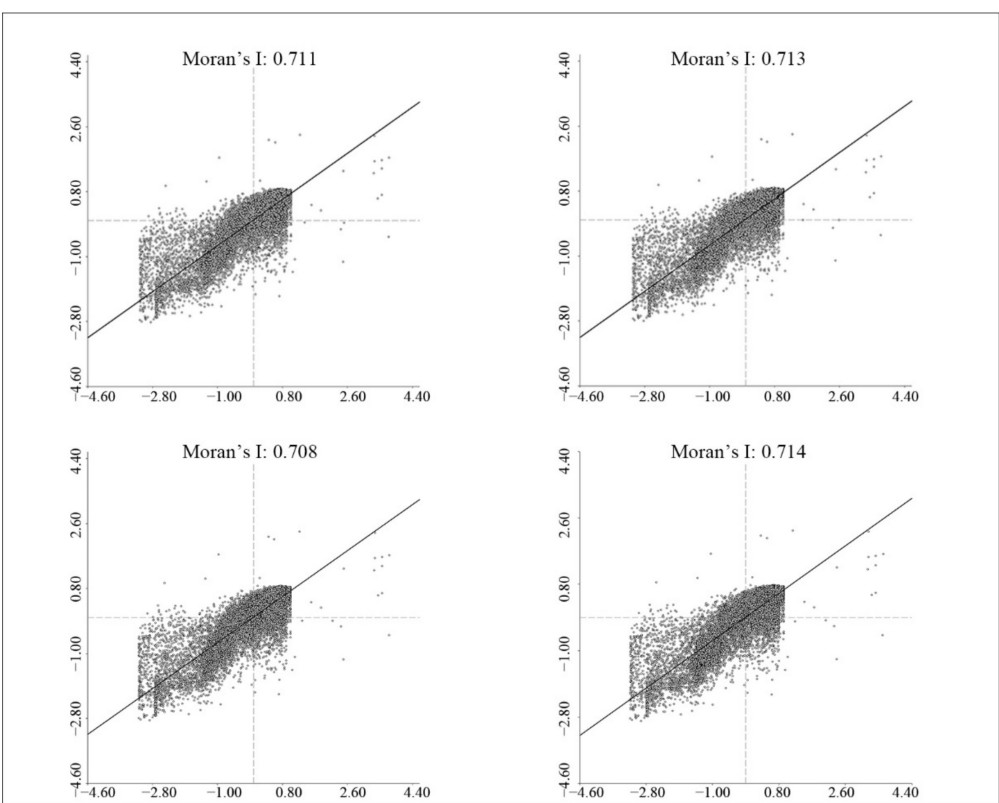

**Figure 9.** Moran scatter charts of the carbon storage in Hangzhou under the four scenarios in 2030.

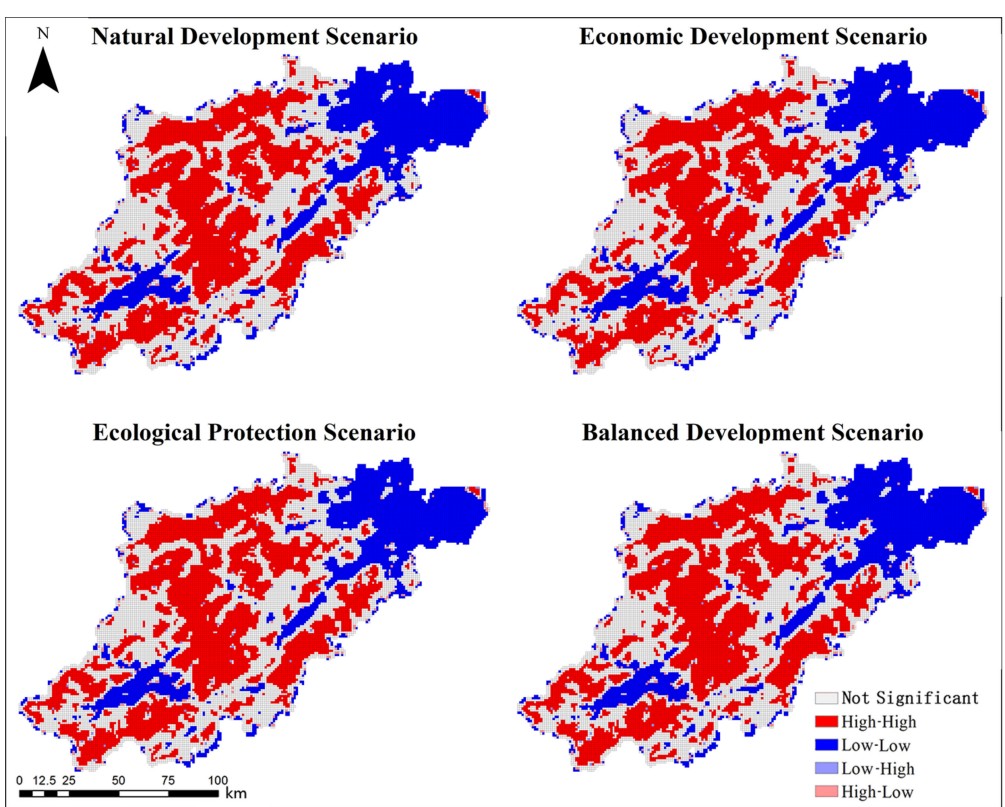

**Figure 10.** LISA agglomeration diagrams of the carbon storage in Hangzhou under the four scenarios in 2030.

### 4.7. Zoning of Carbon Storage in 2030

Using the natural breakpoint method in ArcGIS 10.2, the spatial distribution map of Hangzhou carbon storage in 2030 under the four scenarios was mapped and divided into a lower-carbon-storage area, low-carbon-storage area, medium-carbon-storage area, high-carbon-storage area, and higher-carbon-storage area (Figure 11). The results showed that the overall characteristics and patterns of carbon storage values in Hangzhou in 2030 under the four scenarios remained unchanged.

The lower-carbon-storage area and low-carbon-storage area were mainly distributed in the northeastern and southwestern areas of Hangzhou, and control measures could be formulated for this area in the future to avoid further declines in the carbon storage value. The higher-carbon-storage area and high-carbon-storage area were mainly distributed in the northern, central, southern, and southeastern areas of the city in which more ecological land covers should be protected.

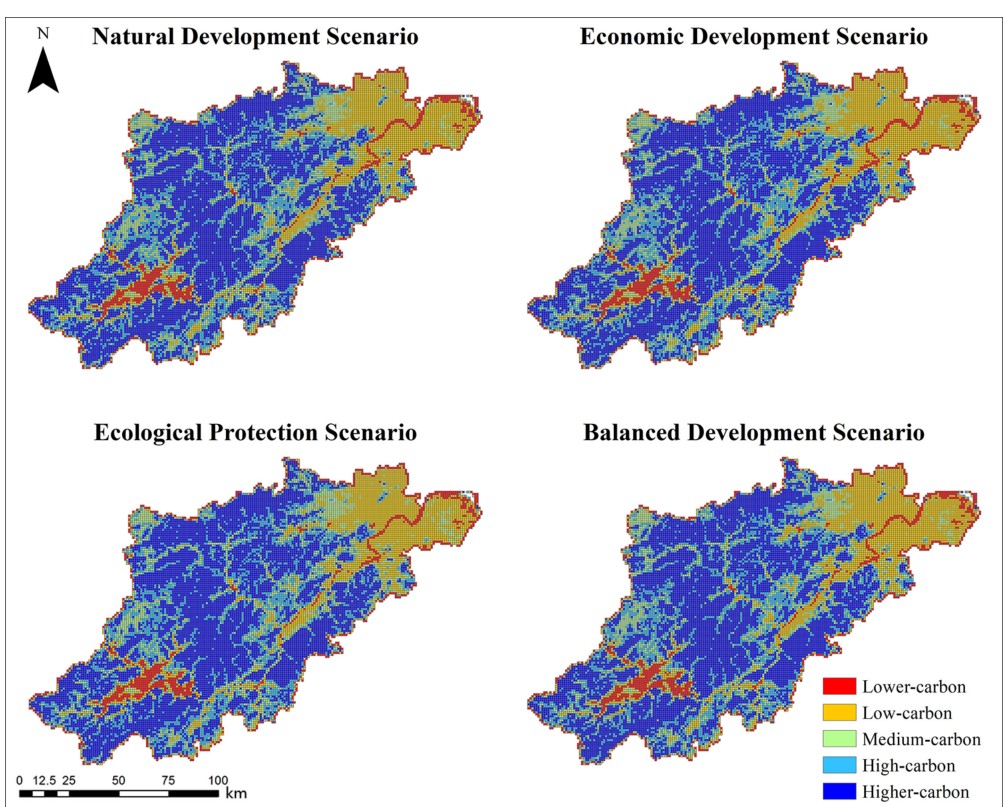

**Figure 11.** Carbon storage zoning in Hangzhou under the four scenarios in 2030.

## 5. Discussion

*5.1. Comparative Discussion and Analysis of the Four Scenarios*

According to the results obtained from the simulations, compared with 2020, the carbon storage in Hangzhou will further decrease in all four scenarios in 2030, which will be due to the increase in the area of artificial surfaces and the decrease in the area of cultivated land.

In terms of specific scenarios, under the ND scenario, the change in Hangzhou's carbon storage will continue the trend of 2000–2020, i.e., with population and economic growth, there will be further expansion of artificial surfaces and a decline in forest and cultivated land. Under the ED scenario, the decline in Hangzhou's carbon storage will accelerate due to the further increase in the expansion of artificial surfaces. Under the EP scenario, Hangzhou's carbon storage will slow down significantly, decreasing by only 0.62%, due to the implementation of strategies to strictly protect forests, wetlands, and water bodies, and strictly limit the growth rate of artificial surfaces. This scenario is consistent with building an ecological city and a low-carbon city in Hangzhou and is worthy of further study. Finally, under the BD scenario, the carbon storage in Hangzhou will be lower than in the ND scenario and only slightly higher than in the ED scenario because of the reduced restriction on the growth in artificial surfaces. The economic and ecological benefits of this scenario will both increase compared with the natural development scenario. This scenario will be in line with the objectives of constructing an ecological city and economic development in Hangzhou and is also worthy of further extension studies.

In summary, among the four scenarios, both the EP and BD scenarios will be conducive to Hangzhou achieving several planning objectives and should be further studied in the future to achieve optimal ecological and economic benefits and the highest carbon storage capacity.

*5.2. Possible Directions for Improvement*

This study applied a framework that combined MOP, the PLUS model, and the InVEST model to predict carbon storage in Hangzhou in 2030 under four different scenarios. However, the framework used in this study still has shortcomings.

First, in simulating the spatial distribution of land cover in Hangzhou in 2030, the Markov chains modules of the PLUS model and MOP were used to predict the demand of each land-cover type under different scenarios; however, Hangzhou has a large water network and future global warming may lead to a reduction in the area of water systems, especially in Qiandao Lake and Yuhangtang River. However, in the PLUS model, it is not possible to reflect the potential impact in this respect. Therefore, in the future, when modeling the spatial distribution of land cover, future climate change should also be taken into account, as well as human activities and policies.

Second, in predicting the carbon storage in Hangzhou in 2030, this study used the InVEST model to measure the carbon storage, where the carbon density data were obtained by reviewing the previous literature. However, the InVEST model ignores both differences in vegetation type and growth conditions for the same land type and differences in carbon density values over time. Therefore, in the future, when measuring carbon storage, a finer division of land classes is needed and large-scale soil sampling should be used to obtain accurate carbon density data.

## 6. Conclusions

Studies of LUCC and carbon storage have been conducted globally in a wide range of situations. However, despite a considerable amount of the extant literature using land prediction models to simulate LUCC and carbon storage changes under different scenarios, few scholars have considered economic and social benefit markers when establishing the simulation scenarios, nor have many explored fine-grained zoning based on carbon storage predictions. We proposed an integrated framework that couples MOP, the PLUS model, and the InVEST model to simulate and predict the spatial distribution of land cover and carbon storage under different scenarios, taking into account multiple objectives of ecological conservation and economic development. The following conclusions were obtained. From 2000 to 2020, the areas of artificial surfaces and wetlands in Hangzhou grew rapidly, whereas the area of cultivated land decreased significantly. Carbon storage showed an accelerated decline, with a cumulative decrease of $5.7 \times 10^6$ t. Moreover, the simulation results of land cover in Hangzhou under four different scenarios showed that the areas of artificial surfaces, water bodies, and shrubland will continue to increase, whereas the areas of cultivated land and grassland will continue to decrease by 2030 compared with 2020. The ED scenario predicted the most rapid expansion of artificial surfaces, with an area of 2038.00 km$^2$, whereas the EB scenario predicted the smallest area of artificial surfaces at 1667.46 hm$^2$. Compared with the ND scenario, the BD scenario had 4.91% more economic benefits and 1.18% more ecological benefits, which can effectively achieve the multiple objectives of ecological protection and economic development. The predictions of carbon storage in Hangzhou under four different scenarios showed that the carbon storage will continue to decline up to 2030 compared with 2020, with the areas of decline mainly concentrated in Yuhang, Fuyang, Xihu, Xiaoshan, and Jianggan. The ED scenario will result in the lowest carbon storage at $228.67 \times 10^6$ t, while the EB scenario will have the highest carbon storage at $232.23 \times 10^6$ t. This scenario will strictly limit the expansion of artificial surfaces and prohibit the reduction in the total amount area of water bodies, wetlands, and forests, which will effectively slow down the decline in carbon storage. Finally, the results of the spatial correlation analysis and zoning of carbon storage in Hangzhou under the four different scenarios showed that the spatial distribution of carbon storage was inextricably linked to the land cover, exhibiting more obvious characteristics of high–high concentration and low–low concentration. The zoning of carbon storage can help decision makers in formulating different conservation and control strategies.

Most cities in the Yangtze River Delta region, such as Hangzhou, are facing multiple objectives of economic development, ecological protection, and low-carbon development; therefore, the presented case study of Hangzhou can represent a reference in the following aspects: (1) the framework of MOP–PLUS–InVEST can quantify the results of urban land-cover change under different scenarios from three aspects: economic benefits, ecological benefits, and carbon storage; (2) the application of carbon storage zoning can combine urban carbon storage management with future urban zoning management, which can help managers to adopt different planning strategies for different zones and help to achieve the "double carbon" goal of urban metropolitan development.

Although meaningful results and findings were revealed through this study, other issues need to be explored and discussed in further research. The achievement of carbon neutrality and peak carbon targets requires attention, not only regarding carbon storage but also carbon emissions. However, this study only discussed carbon storage, which may not be sufficient for policy support; thus, future research needs to integrate carbon storage and emissions to provide data and policy recommendations for achieving carbon neutrality targets. In addition, this study only explored the relationship between urban land cover and carbon storage in different contexts at a single scale; thus, future research needs to quantify and compare the impact of urban land cover on carbon storage from a multi-scale perspective.

**Author Contributions:** Conceptualization, Y.L. and S.Y.; data curation, H.J., H.W. and Q.R.; formal analysis, Y.L., S.Y., X.G. and D.G; funding acquisition, Y.L. and D.G; methodology, Y.L. and S.Y.; project administration, Y.L. and D.G.; resources, Y.L. and D.G.; software, S.Y., H.J., H.W., X.G. and X.D.; supervision, Y.L. and D.G.; validation, Q.R., X.G. and X.D.; visualization, H.J., H.W. and Q.R.; writing—original draft preparation, Y.L., S.Y., H.J and H.W.; writing—review and editing, Y.L., S.Y., H.J., H.W. and D.G. All authors read and agreed to the published version of the manuscript.

**Funding:** This research was funded by the National Natural Science Foundation of China (Grant No.51878593), and the Center for Balance Architecture, Zhejiang University (Grant No.KH-20212946).

**Institutional Review Board Statement:** Not applicable.

**Informed Consent Statement:** Not applicable.

**Data Availability Statement:** All the data used for the study appear in Section 2.2 of the submitted article.

**Acknowledgments:** This research was supported by the National Natural Science Foundation of China (Grant No.51878593), and the Center for Balance Architecture, Zhejiang University (Grant No.KH-20212946).

**Conflicts of Interest:** The authors declare no conflict of interest.

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
