# Peer review of "Spatial-Temporal Evolution and Prediction of Carbon Storage: An Integrated Framework Based on the MOP–PLUS–InVEST Model and an Applied Case Study in Hangzhou, East China"

_land, doi:10.3390/land11122213_

Round 1

Reviewer 1 Report

This is a place-based, rigorous research on spatial-temporal evolution and prediction of carbon storage in Hangzhou, using multi-source data. The topic is prevailing and its main contributions may be embodied in simulation scenarios of carbon storage under differed planning scenarios. To justify this contribution,the authors could detail a bit more about table 6, on how carbon storage is calculated, which is the foundation for follow up prediction and simulation. As there isn't a section for review, this possibly can be done in introduction or method, spending some contents on how existing references inform this study. Currently, what have been done before or what previous research tell is somehow insufficient. 

The efforts of fine grained examination in Hangzhou are appreciated, yet readers may expect to see how study at the fine grained level can imply for other places beyond Hangzhou. Apart from the place-based, rigorous study on the specific site, which is likely to be promoted to other similar cities, what can be generalized from this research. 

Minor issues, e.g., table 5 is not very clear in terms of format. Some of the  figures can hardly tell the difference, may be highlight the difference between in for example Figure 10-12. 

Reviewer 2 Report

Land-2042585_14/11/2022

Review: Spatial-Temporal Evolution and Prediction of Carbon Storage Based on MOP-PLUS-InVEST Model: A Case Study in Hangzhou, China 

First, I would like to express my congratulations for touching on an issue, namely the relation of land use/cover change to carbon storage for environmental protection and sustainable development on a regional scale. However, a scientific discussion of the results must be improved in the manuscript. Moreover, the quality of the manuscript is average and should be improved. Thus, some considerable corrections are suggested:

1.     Broad text editing has to be done as there exist many mistakes not only in the abstract.

2.     The title should be corrected as research proposes “an integrated framework” based on MOP and two models (see lines 75-77)

3.     Several terms used should be explained, e.g. ecological protection, ecological conservation, natural development, territorial spatial planning, GDP density, carbon density values, or use explicit references to proper sources. The same regarding abbreviations, e.g. DEM, NDVI, GM.

4.     The last section in the Introduction should be clarified or divided (see lines 75-81). The meaning of references [27-29] is not understood there as well.

5.     Why Tables 6-8 are named the same? What is the meaning of unit: t/hm2 there and in Table 10? Please, avoid the reader’s confusion about Area/km2 in Table 7.

6.     Section 5. Discussion must be improved as it clearly demonstrates either unfinished study or works to be done in the future!? Perhaps, a scientific discussion of the results could be added to section 4. Conclusions (section 6) has to include suggestions for further research at the end.

7.     At least reference 27 has to be corrected.

In view of all the above-mentioned issues, I recommend to the editors that MAJOR REVISION is addressed before accepting the paper for publishing.

Wishing you the best of luck with your research, and thanking you again for conducting this study.

With best regards

The reviewer

Reviewer 3 Report

Indeed, the problem of the article is very important for China.
I suggest adding the addition to the title that it's about East China?
Figure 1 is not cited. Please check the others.
There is no short geographical description of the study area
Figure 2 Is illegible. I can turn it. Vertical layout.

The map (fig. 3) with land use is too small. You can make one legend for it. Similarly Figures 6 et 12.

Round 2

Reviewer 2 Report

Dear Editors and Authors

The revision seems properly addressed my recommendations. I have no further suggestions, and the paper can be considered for publishing.

Wishing you the best of luck with your research, and thanking you again for conducting this interesting discussion.

With best regards

The reviewer

Reviewer 3 Report

I accept corrections